# Changing Perceptions and Uses of "Companion Animal" Public and Pseudo-Public Spaces in Cities during COVID-19 Pandemic: The Case of Beijing

Haoxian Cai 🆔 and Wei Duan *🆔

School of Landscape Architecture, Beijing Forestry University, Beijing 100083, China
* Correspondence: duanwei@bjfu.edu.cn

**Abstract:** This paper examines the debate over the place of "companion animal" public space in China's cities. With the COVID-19 outbreak, this debate has entered a new phase, where the social response to the outbreak may have fundamentally changed the public's use and perception of "companion animal" public and pseudo-public space. This paper combines quantitative and qualitative analysis of posts and comments on two of China's largest social media platforms with a big data approach, based on a case study in Beijing, China. There were statistically significant differences in the perception and use of "companion animal" public spaces and pseudo-public spaces before and after the pandemic. We attribute the impact of the pandemic on "companion animal" spaces to three pathways: changes in opportunity, changes in ability, and changes in motivation. We found that the pandemic led to an increase in the amount of time available to some people but a decrease in the amount of "companion animal" public space available due to the pandemic closure. In addition, the use of "companion animal" public spaces in pseudo-public spaces declined, while those located within the open urban green space on the city's outskirts stood out after the outbreak. With the normalisation of the pandemic, there will be new challenges for the development and operation of companion-animal-related public spaces in cities, which will be the next focus of research. In addition, governments and social media should work together to promote and support sustainable animal ethical practices to better respond to the crisis. These findings will help complement the urban services system and guide future planning, design, and evaluation of related spaces.

**Keywords:** public space; privatisation; companion animals; animal ethics; China; COVID-19

## 1. Introduction

Cities, as spaces where high densities of people and goods congregate, provide an important channel for spreading infectious diseases. Urbanisation promotes spatial overlap between hosts within vectors, which facilitates the rapid spread of pathogens [1], as recently shown by COVID-19. At least 60% of newly developing infectious illnesses are thought to be spread from wild to domesticated animals and people [2]. According to recent research investigations, it is likely that COVID-19 originates from zoonotic diseases [3]. In complex urban systems, urban livestock and pet-keeping practices, the mobility of animals in urban spaces, and the direct impact of urbanisation on their physical environment become driving forces that may generate diverse transmission chains at the wildlife–domestic animal–human interface [1,4]. Among these, urban companion animals, the leading domestic animal species in urban spaces, have essential links and critical functions in the pandemic transmission interface. In contrast to rural regions, companion animals are completely included into family life in urban settings, where their living circumstances, such as free range and frequent outdoor activities, may result in intimate encounters between human and urban wildlife [5]. As a result, there is a renewed social debate about urban companion animals [6,7].

The study of companion animals In urban spaces has a long history in academia. Since the development of zoogeography, the question of human–animal interactions in cities and how different urban spaces shape complex, mutually constructed human–companion animal relationships have been important research topics [8,9]. In the literature related to humans and companion animals, in public spaces, in particular, dogs have been a very important area of research [10], with research currently focusing on discussions around the dichotomous relationship between human and companion animal oppositions in urban spaces and the contradiction between better integrating dogs into social life and regulating their rights in urban space [6,11].

At the same time, debates about the end of public space have gradually come to the forefront with the rise of urban privatisation [12]. The most controversial of these is privately owned public open space (POPOS) or private public space (POPS), as opposed to publicly owned open spaces, such as parks and squares, which the government traditionally provides; it is an outdoor or semi-outdoor space on private land or private property, built and managed by private investment in exchange for public use through government incentives [12]. Therefore, some scholars have begun to examine the impact of the privatisation of urban public space on companion animal space, mainly from a politics of rights perspective. For example, Sue Donaldson has repeatedly discussed the negative impact of the privatisation of public space on animal rights, highlighting the significant practical and conceptual challenges facing "companion animal" public space [13]. Marie Carmen argues that the privatisation of public space has given capitalist corporations free reign over urban space to maximise their profits and infringe on the rights of companion animals [14]. "Companion animal" public space is a relatively new but growing area of urban research. In many countries, urban planners are beginning to incorporate this companion animal element into land use decisions, with "companion animal" public space being one of these [6]. In the United States, dog parks are common as "companion animal" public spaces in cities across the country [10,15]. However, most cities in China lack dedicated companion animal parks for humans to interact with their companion dogs, and thus the relevant spaces in China are all rather vague.

The shift from an agrarian to an industrial, post-industrial, digital society has primarily influenced the perception of companion animal spaces as "companion animal" public spaces. However, the COVID-19 outbreak seems to have been a turning point, with the government setting restrictions to limit social gatherings and crowding and to avoid contagion. People have experienced a degree of lockdown, and society has changed significantly in various ways [16,17]. The COVID-19 outbreak is likely to affect the perception and use of "companion animal" public spaces. The possible impact of COVID-19 on public space has been discussed in the literature [18–21]. However, there is still a lack of empirical research on the "companion animal" public space in China. Thus, in the Chinese context that is experiencing a new type of urbanism very different from that of the West [22,23], we based our social media data on Weibo and Xiaohongshu, using content analysis methods in conjunction with natural language (NLP) analysis and GIS spatial analysis to investigate the extent to which the epidemic will change the way people perceive and use "companion animal" public spaces and pseudo-public spaces. At the same time, we conducted semi-structured in-depth interviews to support the validation and correction of the big data findings in order to conduct a more in-depth study. The results of this study will help to complement the functional system of urban services and guide the planning, design, and evaluation of related spaces in the future.

## 2. Materials and Methods

Beijing is one of China's most crowded and dynamic metropolises, with a relatively well-developed public space infrastructure. Then, as one of the most thriving real-estate markets in China, property values are well above the national average, with shopping centres and commercial complexes ranked in the top tier in China. High property values lead to gentrification, and there is a strong need for social isolation and spatial control of

the city's pseudo-public spaces. In addition, as Beijing is the capital, it is also representative of the management of the pandemic. Thus, by choosing this city, we can better investigate the effectiveness of urban governance of public and pseudo-public spaces for "companion animals" during the outbreak.

As the study was conducted on "companion animals" in public and pseudo-public spaces, urban companion animals do not have language skills, so we turned to the owners who are closest to their companion animals and reflect their needs, as well as the public who have a close relationship with them. According to the China Pet Industry White Paper 2021, the number of pets in China is predicted to reach 220 million in 2022, with dogs accounting for the largest share. Therefore, we decided to study the pet dogs that are kept in the largest numbers and most studied in the literature. We first ruled out the questionnaire approach due to its inherent reliance on participant responses, as there may be issues of recall and social desirability bias [24]. In addition, the information received from participants is difficult to verify independently, especially for green space surveys in the parkland category, where there is a large margin of error [25]. There are also significant limitations to direct in situ observations, as samples outside of a given observation time cannot be reliably estimated, thus requiring multiple observations over different days and seasons to ensure reliability [26], and thus direct observation studies that require significant time and often lack longitudinal depth and breadth. At the same time, there are certain shortcomings in big data methods of measurement, which can be well compensated for by in-depth interviews [27]. In addition, Flick et al. mention that respondents' views are more likely to be expressed in a reasonably open design rather than in a standardised questionnaire [28]. Therefore, we selected a method of social media big data combined with semi-structured in-depth interviews to investigate. The method we chose has three advantages.

Firstly, our comprehensive research methodology allows us to collect a wealth of data. Social media data mining allows us to collect much larger volumes of data than traditional field research, and computational profiling of the data helps us to perceive better and identify the changing focus of urban perceptions and uses of public and pseudo-public spaces for "companion animals". Our analysis is based on verbatim and textual big data analysis, which allows us to fully reflect the media landscape on the subject of urban companion animals and urban multi-species interactions and relationships. The flexibility and adaptability of in-depth interviews allows us to obtain more in-depth information and evidence, and to explore more perspectives, layers, and dimensions of the issues uncovered by social media data.

Second, the study data we used are reliable and general. Due to the internet's ongoing development over the past few years, it has integrated into people's lives. The fast growth of social media and the widespread use of smartphones have created more and more well-liked and respected platforms for individuals to openly express their thoughts. New methods for comprehending the traits of social activity are made available by the large volume of data on social media. Textual data from social media create a large database of public impressions, including information that is challenging to obtain through conventional polls. In addition, the atmosphere on the internet is more relaxed and less morally constrained, which better reflects the true psychological state of the interviewees. In-depth interviews, although they may be subject to greater ethical and identity constraints, allow for more in-depth and repeated discussion of certain topics and an understanding of the reasons behind them. Fitting the results from the in-depth interviews to the big data, thus, allows for a high degree of accuracy and generalisability of the data.

Thirdly, social media data combined with a semi-structured in-depth interview approach have allowed us to reflect public perceptions during the outbreak accurately. The Chinese central government encouraged the populace to limit their exposure to public settings after the COVID-19 outbreak. Provincial and local governments have proceeded to create more stringent community access management measures in accordance with the central government's epidemic prevention plans, mandating the populace to remain indoors in order to further stop the spread of the virus. Popular social media has consequently evolved

into the fundamental platforms for individuals to learn about the progression of the disease and share their opinions. Therefore, the greatest approach to comprehend public thoughts and attitudes during an outbreak is through textual data from social media. Additionally, because social media data are instantaneous, they may give quick and efficient feedback on shifts in perception as the pandemic develops. At the same time, the semi-structured in-depth interview approach makes up for the shortcomings of big data by providing a great deal of in-depth information compared to the relatively short and fast information conveyed by big data. It also provides a new perspective on public perceptions during the outbreak.

However, it is equally important to recognise the limits of our strategy. Highly diverse social media users occasionally influence and hold the perceptions of the general population, limiting the representativeness of social media data for profiling. In addition, the non-social-media-loving public may introduce more error into the analysis part of our big data study, thus limiting the generalisation of the findings. Moreover, the study is only at a preliminary stage, the scope of the study is limited to Beijing, China, and the number of respondents in the in-depth interviews is limited by the study and, thus, the sample size is small. In addition, the overall perspective of the study is from a macro perspective, and future research is needed to combine more methods for multiple perspectives and further segmentation.

Our research methodology was divided into primary data collection, data processing, data analysis (including natural language (NLP) analysis and spatial and content analysis), and semi-structured in-depth interviews, as shown in Figure 1. To collect data for the study, we used a Python web crawler to search the original text posted by Weibo and Xiaohongshu users from 1 May 2018 to 1 May 2022, restricting the search to Beijing, China, based on the two Chinese keywords "dog walking space" and "good places to walk your dog", with the time interval divided into before (January 2020) and after the outbreak, as shown in Appendix A Figure A1. We save the crawled blog data to a local server as the main data source. Weibo is one of the most important social media platforms in China. Similar to Twitter in its powerful interactive features and timely information updates, it has a significant impact on the organisation of social life and public opinion. In the textual resources of Weibo, key and commonly used words can reflect various public narratives and the extent to which the public pays attention to these narratives [29]. Xiaohongshu (Little Red Book), a lifestyle platform and consumer decision portal, can effectively complement and corroborate the Weibo data with its "place seeding" and how-to reviews. Our next step was to clean up the data by removing duplicate content and ads. We obtained a sample of 26,550 valid blog posts, of which 13,150 were posted before the pandemic and 13,400 during the pandemic, totalling over 4.4 million words.

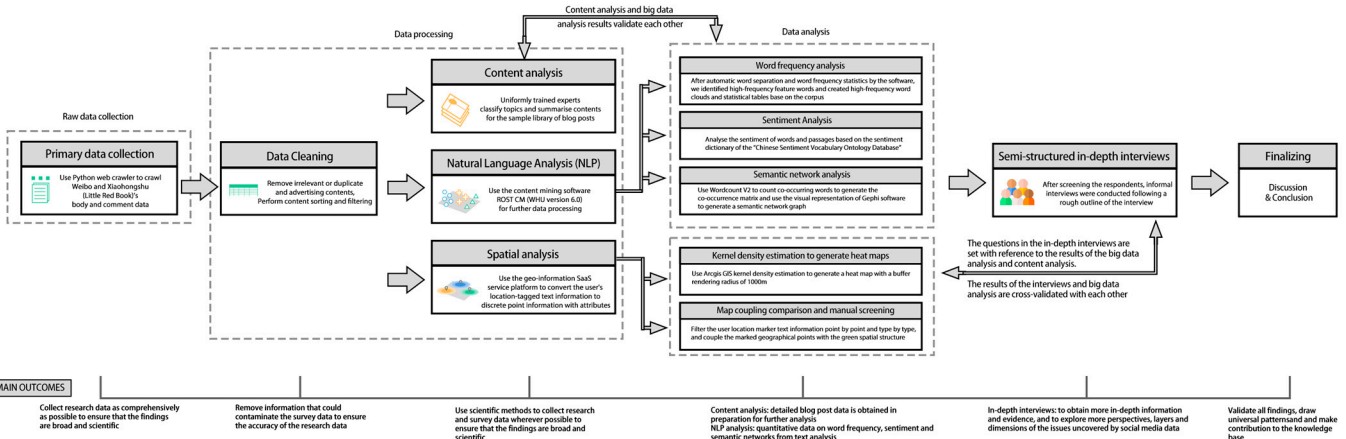

**Figure 1.** Framework diagram of research ideas.

For the data processing, we used the content mining software ROST CM (WHU version 6.0) to perform a natural language processing (NLP) analysis of the acquired textual material. The starting point for the content analysis was to identify the most frequently occurring semantic units in all the text material to provide an overview of potential research topics. Thus, the first NLP analysis we performed was a word frequency analysis. After building a custom lexicon relevant to the research subject, we filtered and segmented the crawled Weibo and Xiaohongshu text data. We removed auxiliary words (e.g., "due to", "this") and merged duplicate words (e.g., "don't" and "can't", "dog", and "hairy child"). High-frequency feature words regarding public perceptions of urban companion animals were identified through automatic software word separation and word frequency statistics. Eventually, we produced perceptual word cloud maps of the most commonly used words in the corpus, as shown in Figures 2 and 3, word frequency statistics for the top 40 words before and during the pandemic, as shown in Table 1 and Word frequency rose charts, shown in Figures 4 and 5.

**Table 1.** Frequency statistics for the top 40 words before and during the pandemic.

| (Before the Outbreak) | | | (During the Outbreak) | | |
|---|---|---|---|---|---|
| **Rank** | **Word** | **Frequency** | **Rank** | **Word** | **Frequency** |
| 1 | Address | 3562 | 1 | Tickets | 3120 |
| 2 | Dog walking | 2819 | 2 | Location | 2916 |
| 3 | Traffic | 2787 | 3 | Parking | 2752 |
| 4 | Parking | 2680 | 4 | Free | 2353 |
| 5 | Tickets | 2235 | 5 | Pandemic | 2073 |
| 6 | Beijing | 1388 | 6 | Dog walking | 1687 |
| 7 | Netizen | 1212 | 7 | Weekends | 1223 |
| 8 | Navigation | 975 | 8 | Camping | 1089 |
| 9 | Kilometres | 971 | 9 | BBQ | 986 |
| 10 | Hours | 863 | 10 | Picnics | 957 |
| 11 | Support | 851 | 11 | Less and less | 935 |
| 12 | Pet friendly | 796 | 12 | Pets | 926 |
| 13 | Camping | 785 | 13 | Beijing | 870 |
| 14 | Location | 754 | 14 | The park is huge | 855 |
| 15 | Place | 735 | 15 | Photo shoots | 823 |
| 16 | Photo-taking | 653 | 16 | Good places to go | 812 |
| 17 | Minutes | 640 | 17 | Cost | 801 |
| 18 | Travel with dogs | 628 | 18 | Recommended | 763 |
| 19 | Today | 617 | 19 | Friendly | 728 |
| 20 | Less crowded | 603 | 20 | Address | 682 |
| 21 | District | 579 | 21 | Playability | 656 |
| 22 | Disadvantages | 562 | 22 | Dogs | 621 |
| 23 | Car journey | 537 | 23 | Blowing wind | 607 |
| 24 | Pet friendly park | 439 | 24 | Lawn | 583 |
| 25 | Tents | 426 | 25 | Recently | 529 |
| 26 | Overall rating | 423 | 26 | Next to | 514 |
| 27 | Kite flying | 415 | 27 | Cute pets | 472 |
| 28 | Fees | 389 | 28 | Minutes | 453 |
| 29 | Shiba Inu | 354 | 29 | Enjoy the flowers | 422 |
| 30 | Free parking | 341 | 30 | Scenic spots | 395 |
| 31 | Dogs can be walked | 339 | 31 | Enjoy the greenery | 354 |
| 32 | Golden retriever | 315 | 32 | Everyone | 351 |
| 33 | Self-drive | 286 | 33 | Around the area | 315 |
| 34 | No entrance fee | 283 | 34 | Suitable for | 278 |
| 35 | Recommended | 274 | 35 | Parking fees | 263 |
| 36 | Route description | 271 | 36 | It's all about the dogs | 251 |
| 37 | Parking lot | 265 | 37 | Leash | 229 |
| 38 | Approximate | 261 | 38 | With hills and water | 217 |
| 39 | Beijing Adoption | 247 | 39 | Specific location | 214 |
| 40 | Opening hours | 239 | 40 | Tents | 208 |

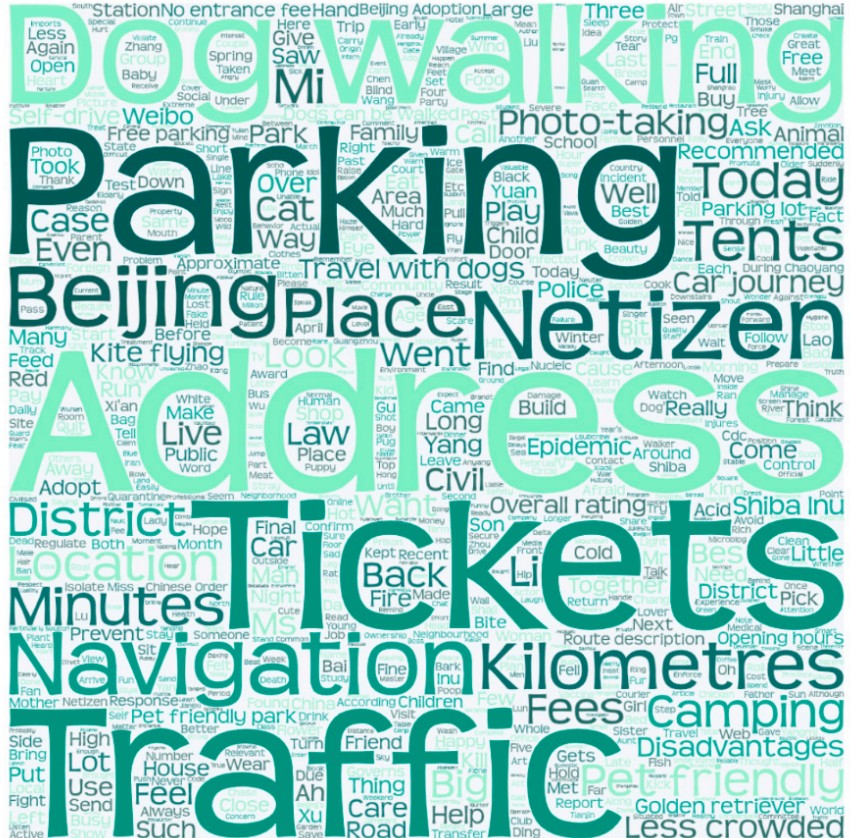

**Figure 2.** Perceptive word cloud map (before the outbreak).

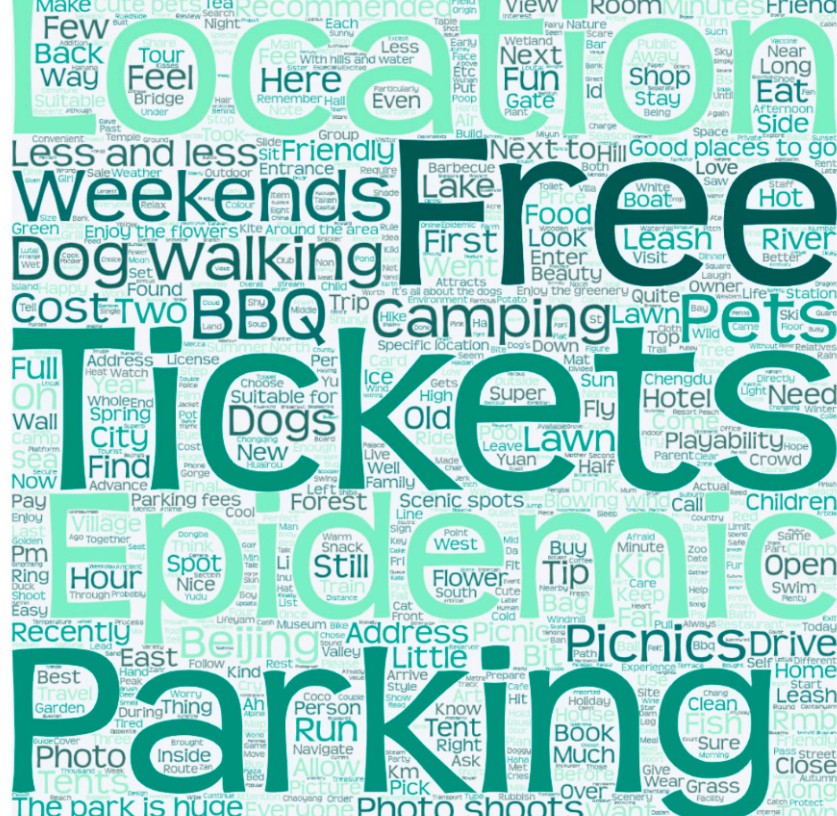

**Figure 3.** Perceptive word cloud map (during the outbreak).

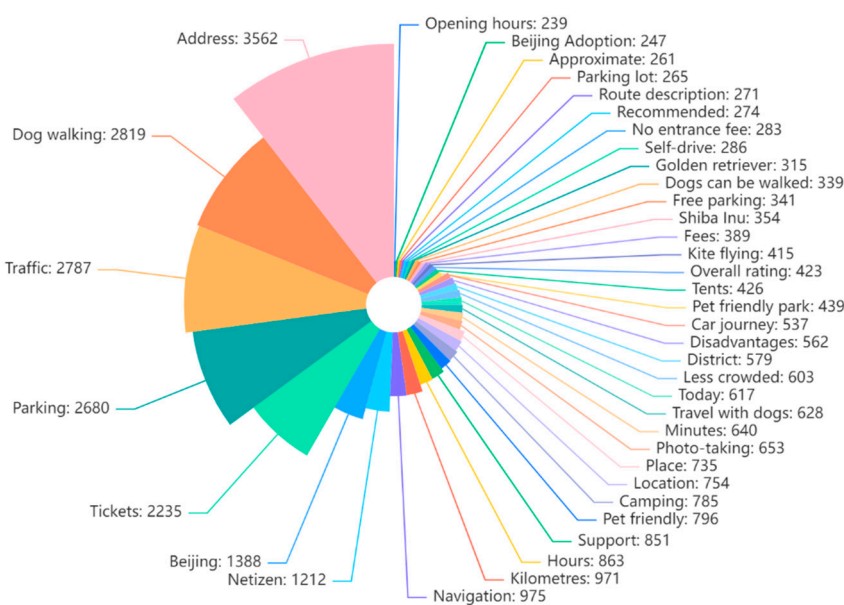

**Figure 4.** Word frequency rose chart (before the outbreak).

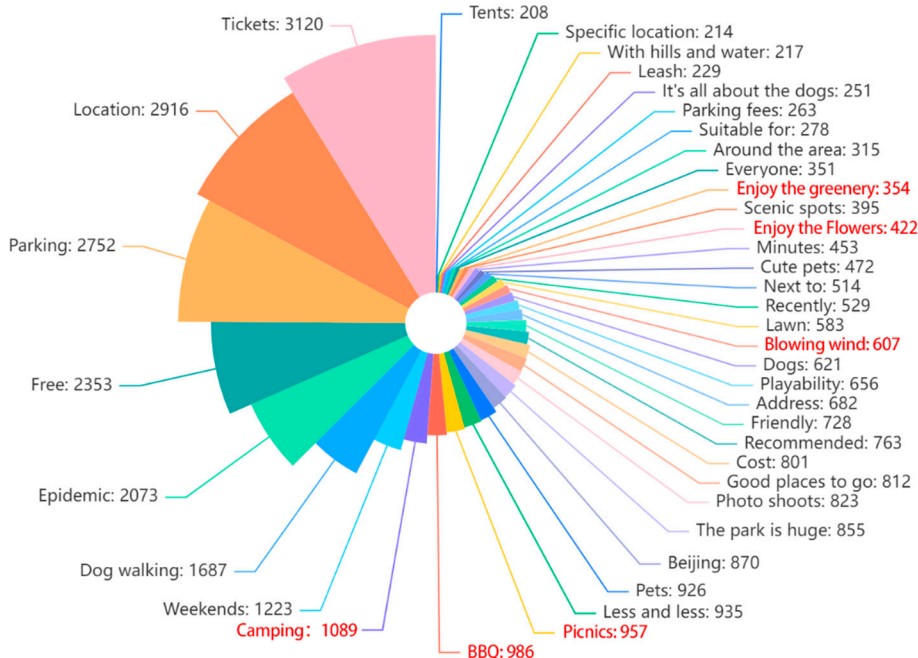

**Figure 5.** Word frequency rose chart (during the outbreak).

After reading the details of the sample of valid blog posts, we analysed the correlations between high-frequency words and used Wordcount V2 to generate a co-occurrence matrix of co-occurring words. Then, we used the visual representation of the Gephi software to generate semantic network diagrams, as shown in Figures 6 and 7. We then performed the sentiment analysis of text data by using a sentiment lexicon based on the Chinese Sentiment Vocabulary Ontology Library to understand the public's general feelings towards companion animal space, as shown in Figures 8 and 9, and Appendix A Figure A2.

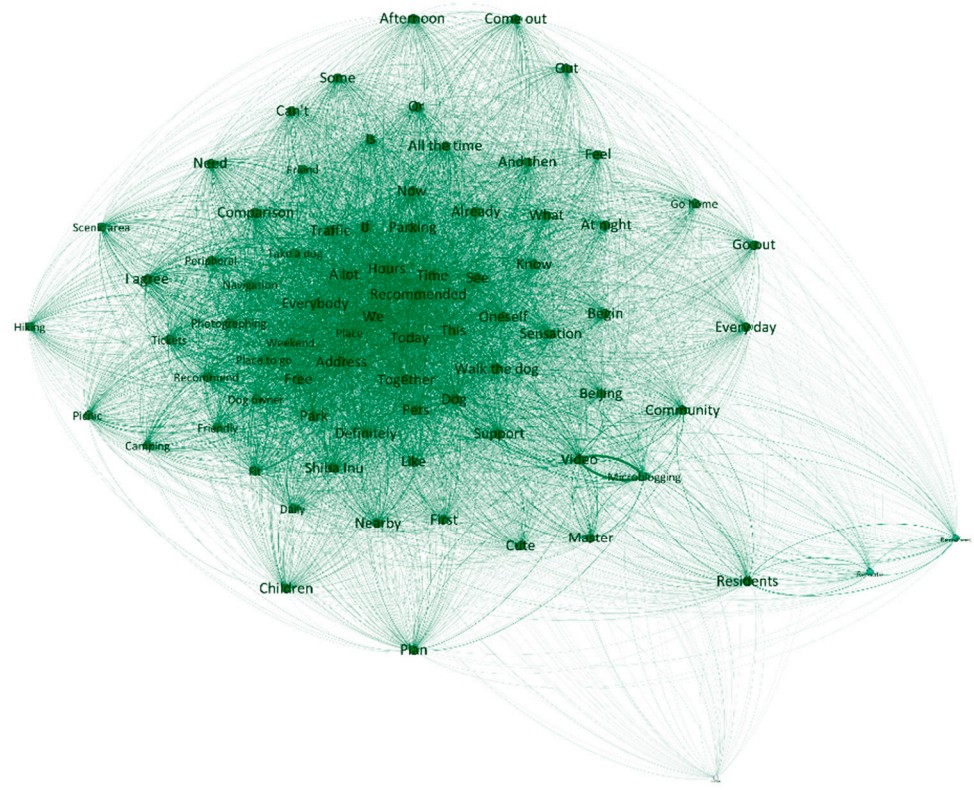

**Figure 6.** Semantic network diagram (before the outbreak).

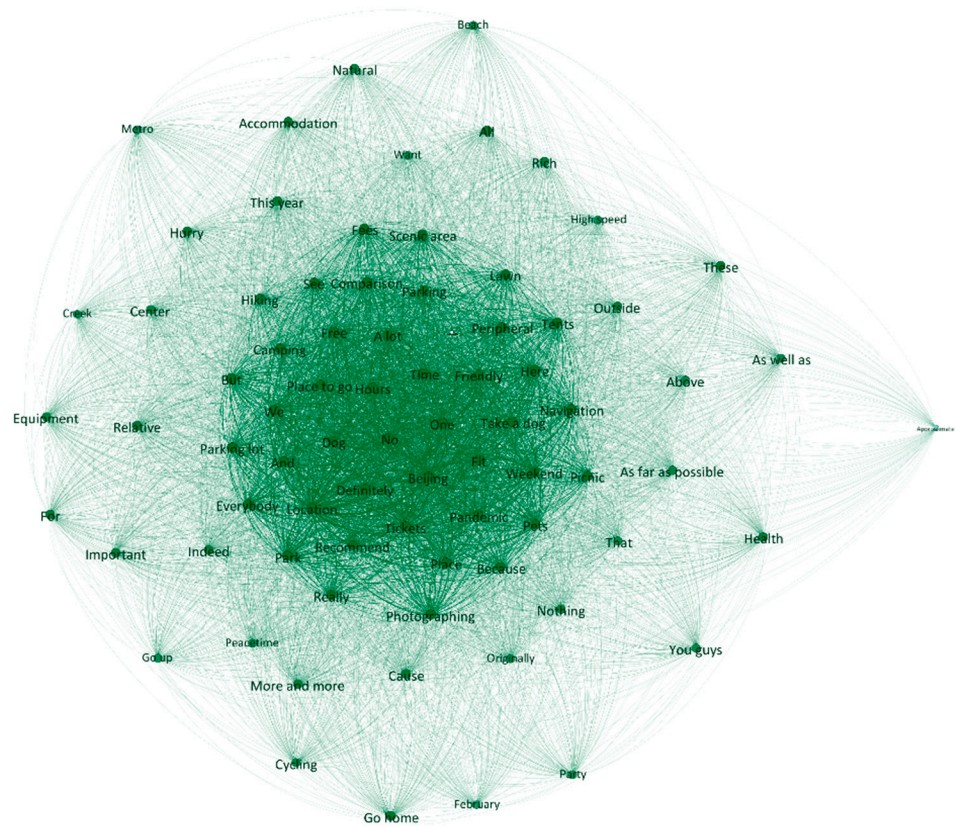

**Figure 7.** Semantic network diagram (during the outbreak).

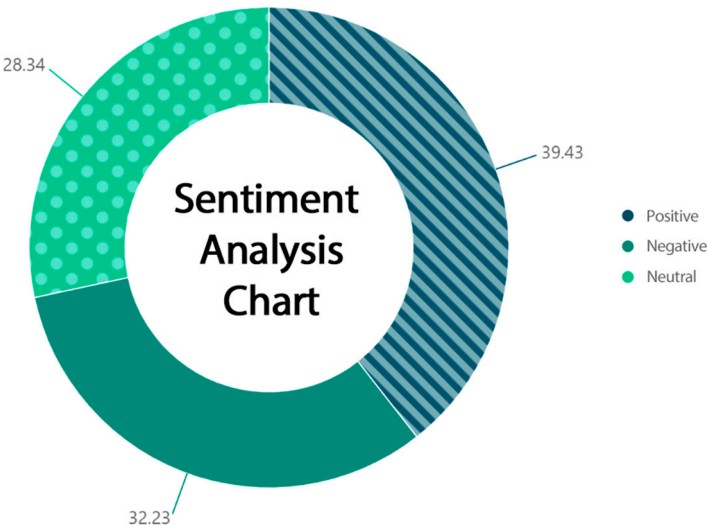

**Figure 8.** Sentiment analysis chart (before the outbreak).

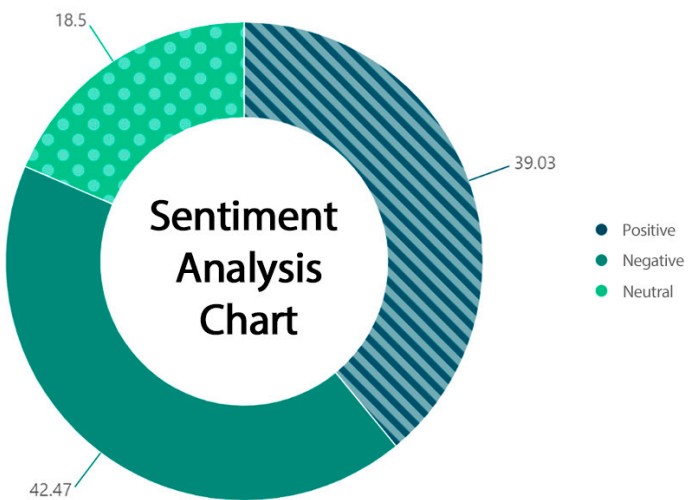

**Figure 9.** Sentiment analysis chart (during the outbreak).

The next step is spatial analysis. We use the Yijingzhilian Geographic Information SaaS platform to convert the user-tagged location text information into discrete point information with attributes, as shown in Figure 10. In addition, to better present the visualisation results, we generated a heat map by Arch GIS kernel density analysis with a buffer rendering radius of 1000 m, as shown in Figure 11. In addition, we also identified, recorded, and analysed the marked points in the pseudo-public space, generating the heat map shown in Figure 12. Geotagging by users on social media is generally motivated by the idea of marking "places worth remembering", which is highly credible and representative [30]. The location of the user's marker reflects, to some extent, the number and spatial distribution of "companion animals" in public spaces. In addition, to further analyse the types of "companion animal" public spaces and improve the accuracy, we conducted a point-by-point screening to identify the classification and coupled the marked geographical points with the green space structure plan of Beijing, as shown in Figure 13.

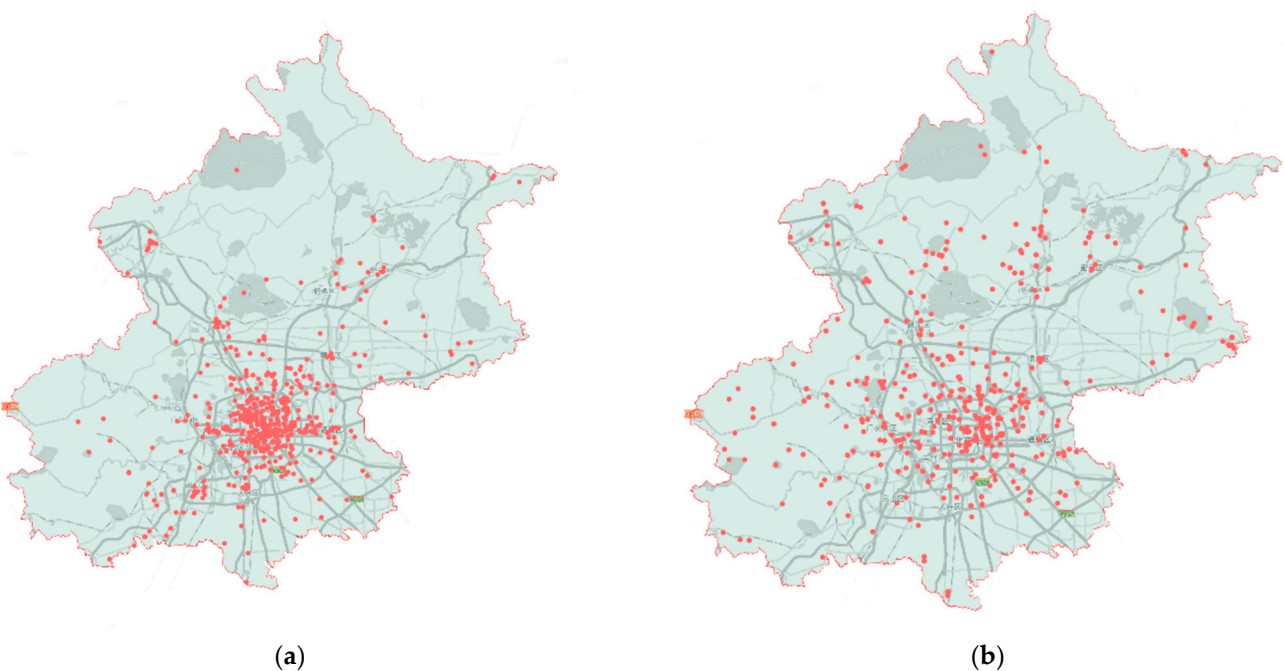

(**a**)  (**b**)

**Figure 10.** (**a**) Geographical distribution of user markers (before the outbreak); (**b**) geographical distribution of user markers (during the outbreak).

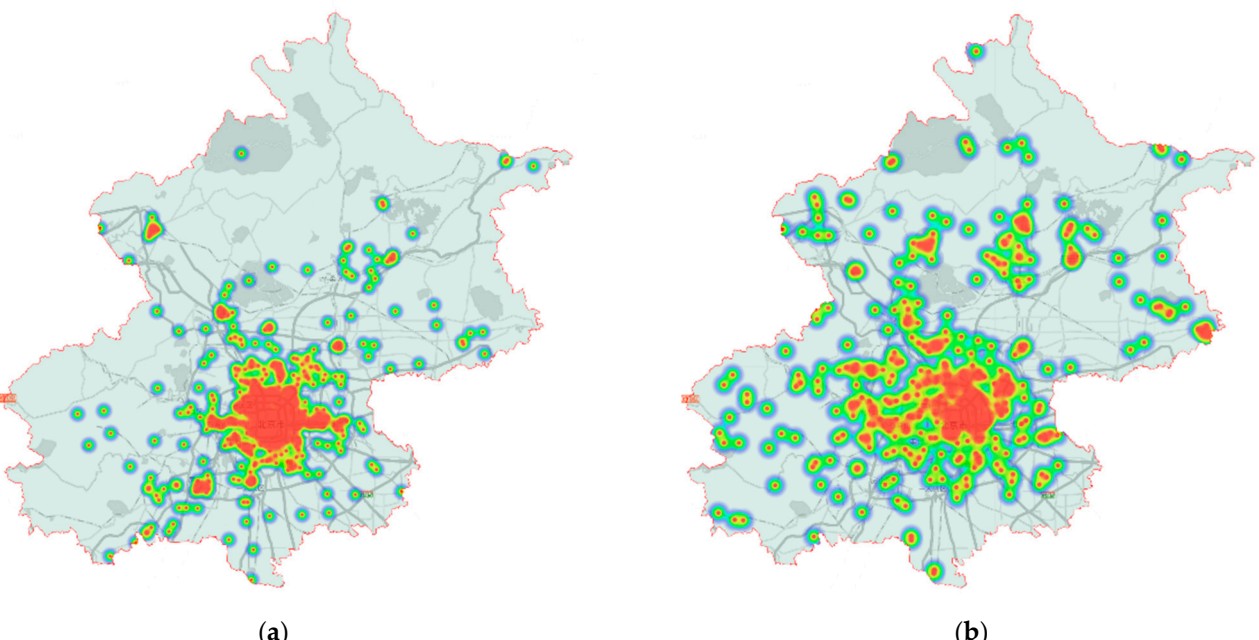

(**a**)  (**b**)

**Figure 11.** (**a**) Heat map of the geographic distribution of user markers (before the outbreak); (**b**) heat map of the geographic distribution of user markers (during the outbreak).

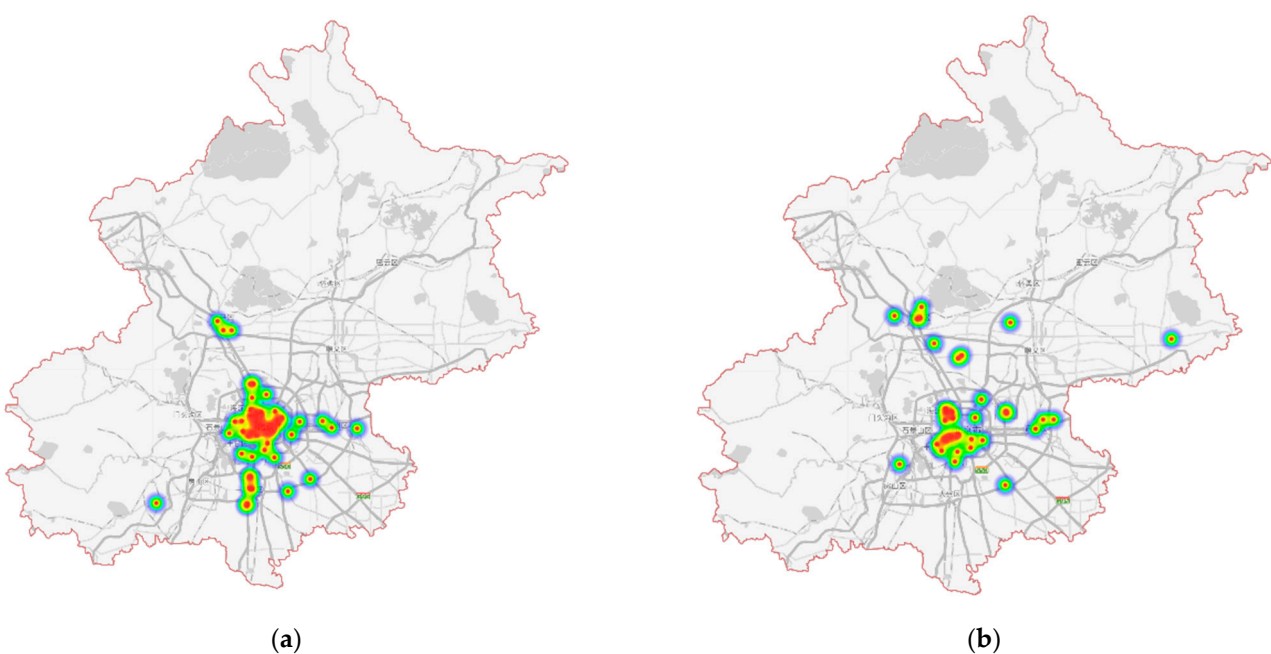

(**a**)                                                                                                (**b**)

**Figure 12.** (**a**) Heat map of the distribution of pseudo-public space markers (before the outbreak); (**b**) heat map of the distribution of pseudo-public space markers (during the outbreak).

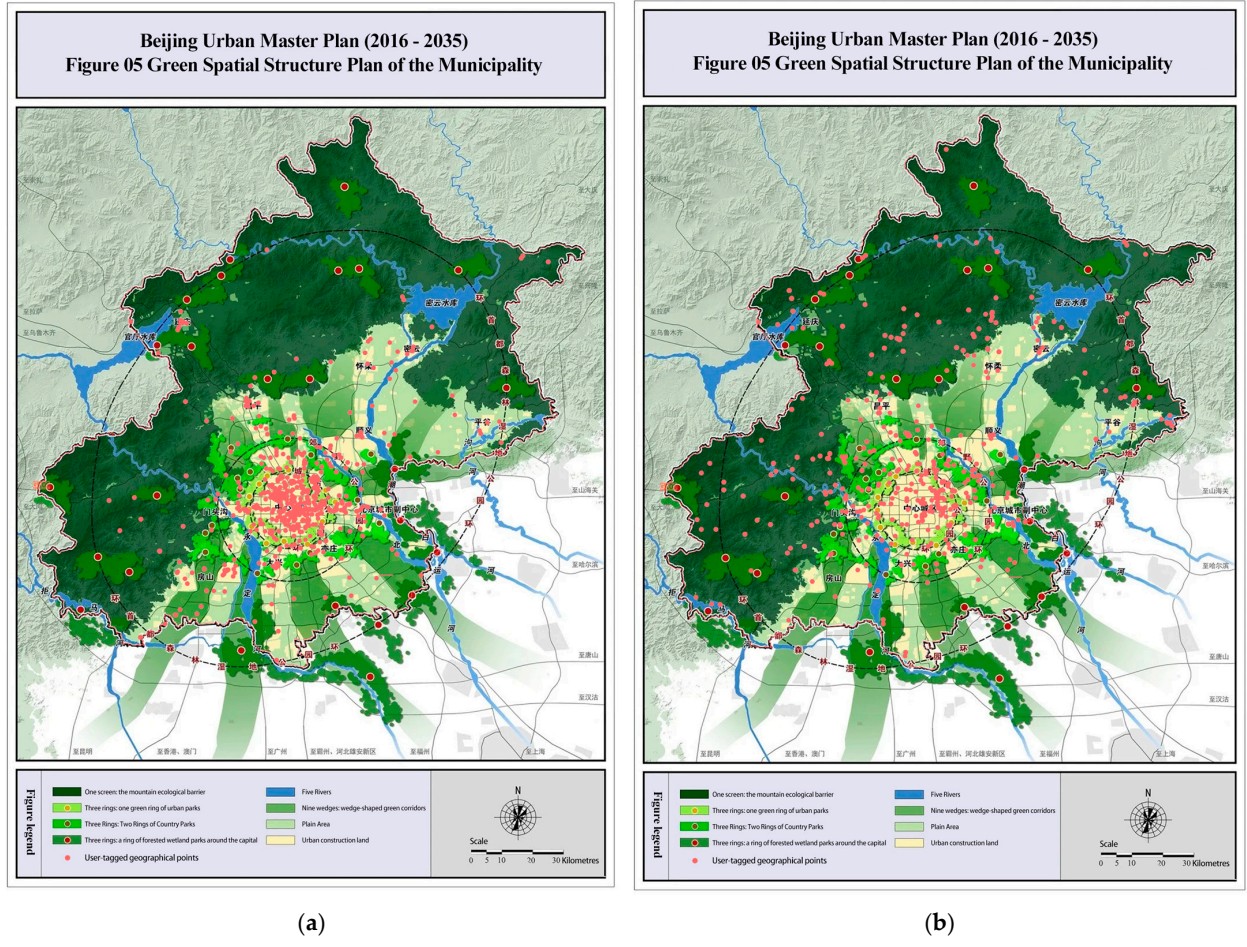

(**a**)                                                                                                (**b**)

**Figure 13.** (**a**) Coupling of user geographic markers and planning maps (before the outbreak); (**b**) coupling of user geographic markers and planning maps (during the outbreak).

Then, a content analysis and topic summary of the sample database of blog posts were conducted by uniformly trained experts. In order to raise accuracy, the topics in the sample database were individually identified by three experts. Cross-referencing was carried out after the identification was completed. In the case of inconsistent themes, experts proofread and discussed until an agreed outcome was reached. We then used the detailed blog post data combined with the results of the big data analysis to interpret relevant social dynamics that reflect public perceptions of the relationship between public and pseudo-public spaces for urban companion animals and COVID-19.

As a final step, we invited respondents to participate in in-depth interviews, Sargeant defined how to ensure the quality of qualitative research participants [31], and we built on his work to select participants who could best inform our research questions and improve our understanding of changes in public perception and use of "companion animal" public spaces and pseudo-public spaces before and after the outbreak. For example, the inclusion criteria for respondents were Beijing residents, having lived with the dogs for more than 6 months before and during the outbreak, being between 18 and 60 years old, etc. Due to the impact of the epidemic containment, we mainly used Tencent Meetings online face-to-face and telephone interviews. This is considered one of the safest methods compared to the restrictions of movement and risk of infection that would result from more traditional face-to-face public surveys. A total of 12 Beijing residents were interviewed in semi-structured in-depth interviews. Consistent with rooting theory, we used a maximum variation sampling strategy to select residents with different lifestyles and conditions, aiming to provide a variety of perspectives. We did not offer any remuneration or incentives to participants. Interviews with respondents ranged from 19 to 33 min in length and were conducted by two interviewers working together in a semi-structured question-and-answer format, divided into an interviewer who was responsible for the interview and an interviewer who was responsible for recording and asking additional questions. We used digital equipment for recording and further transcription and manual thematic coding. All participants gave verbal and written consent to record their interview sessions and followed their views on whether to anonymise their identities. Similar studies, such as C Mayen Huerta et al. [32] and Charlotte Collins et al. [33], have used in-depth interviews to better understand the association between UGS use and other variables. To enhance the scientific validity of the interview questions, we drew, in part, on the Monash Dog Owner Relationship Scale (MDORS) developed by Fleur Dwyer et al. in designing the questions [34]. The use of this scale allows researchers to increase their understanding of human–companion dog relationships by allowing direct comparisons between groups of participants from different demographic or cultural contexts. We also refer, in part, to the Development and Reliability of the Dogs and Physical Activity (DAPA) Tool developed by Hayley E. Cutt et al., which can be used to retest individual, social context, physical environment, and policy-related factors that influence dog owners' dog walking behaviour confidence assessment [35]. The in-depth interview questions consisted of several basic sections, as shown in Appendix A Figure A3, with the questions being fine-tuned to the respondents. The first part collected information about the interviewees and their families, educational level, employment status, and dog status; the second part asked about the general and material impact of the epidemic on the perception and use of the "companion animal" space by the respondents and their pets. In the third part, the focus of the survey was on the mental and psychological effects.

## 3. Results

We derived perceptual word cloud maps before and during the pandemic by performing word frequency analysis on the cleaned social media data for NLP analysis, as shown in Figures 2 and 3. In this case, the frequency of words in the text is proportional to their size. During the COVID-19 outbreak, for the public space of "companion animals", the preoutbreak crawl showed that the most frequently used words were "Address", "Traffic", "Dog walking", "Parking", and "Tickets". We can see that accessibility and ease of parking are the top priorities for people choosing public spaces for companion animals. There is

also much discussion of dog-related topics, such as "Dog hunting", "Chinese Field Dogs", and "Walking dogs without a leash", to name a few popular and common topics.

Surprisingly, the results of the post-pandemic word frequency survey show that words such as "Tickets", "Location", "Parking", and "Free" continue to top the list of words that express dog owners' strong need for "companion animal" public space, while the frequency of activities such as "Camping", "BBQ", "Picnic", and "Blowing wind" and so on has also increased significantly, as shown in Figures 4 and 5. In contrast to public opinion during the pandemic, where public perceptions of urban companion animals tended to be mainly negative experiences [6], for dog owners in Beijing, the word "epidemic" is also at the top of the list, but it is clearly not the main focus of attention. Perhaps as a result of the normalisation of the pandemic, many dog owners tend to see the "epidemic" simply as a social backdrop, focusing mainly on helping their dogs to find new outdoor spaces or as an after-dinner talker, such as "*I can't get out of Beijing because of the epidemic, but I went to Shentang Yu at the weekend, which is one of the few pet-friendly places, and I highly recommend it.*" "*It's rare to see a dog wearing a breathing mask, lol.*" "*Taking Old Man Bear (dog's name) for a walk after lunch, I met Tammy's mother in full armor walking her dog in the yard too!*".

To further explore issues related to public perception, we conducted a semantic network analysis of high-frequency words before and after the pandemic, as shown in Figures 6 and 7. We can easily see that, before and during the pandemic, the semantic network diagram presents a distinct core, basically with several core words as a group expanding outwards diffusely. The hierarchical relationships between the core words are relatively similar. After careful analysis, we identified that "Address", "Traffic", "Park", "Navigation", and "Parking lot" were strongly correlated with each other before the pandemic. Meanwhile, during the pandemic, "Tickets", "Friendly", "Location", "Pandemic", "Parking", "Dog", and "Lawn" have a strong correlation between the words.

In addition, we conducted sentiment analysis on the blog posts before and after the pandemic, mainly analysing the sentiment polarity (i.e., positive, neutral, and negative sentiment) and sentiment intensity of the words with sentiment components within each utterance, and then calculated the total value of each utterance to determine its sentiment category. In order to determine the overall attitude and sentiment tendency of the total opinion data sample, we statistically integrated all statements, as shown in Figures 8 and 11, and found that, before the pandemic, positive sentiment accounted for 39.43% (5283), negative sentiment accounted for 32.23% (4318), and neutral sentiment accounted for 28.34% (3799). During the pandemic, the proportion of positive sentiments was 39.03% (5132), negative sentiments 42.47% (5584), and neutral sentiments 18.5% (2434). We can see that, especially during the pandemic, most bloggers expressed more personal sentiment on the topic of "companion animals" in public spaces, with negative sentiment increasing by 10.24%.

Subsequently, in order to analyse the number and spatial distribution of "companion animal" public spaces, we extracted the geographical locations marked by users from social media data, with a total of 1586 valid points, including 749 before and 837 during the pandemic, as shown in Figure 10. Heat maps were generated to better present the results, as shown in Figure 11. It is clear that the prominent distribution locations were in the urban areas of Beijing, both before and during the pandemic. However, when it comes to identifying the types of "companion animal" public spaces before and after the pandemic, we find that the largest proportion of points is still in scenic areas and parks, with 83% (621) before the pandemic and 91% (761) during the pandemic. In contrast, the remaining points are mainly in pseudo-public spaces, such as shopping plazas and public spaces attached to buildings and certain streets. Comparing the heat maps before and during the pandemic, as shown in Figure 11, it is clear that the density of geographic markers in the built-up area decreases and the density of markers in the surrounding areas increases. After coupling the markers with the Green Spatial Structure Plan in Beijing Urban Master Plan (2016–2035), as shown in Figure 13, we can find that the markers in the peripheral areas are mainly located in parks and green areas, as well as major scenic spots, which coincides with our

differentiation of marker types. Comparing the heat map of pseudo-public space markers, we can also find that the proportion of pseudo-public space has a clear tendency to decrease during the pandemic compared to the pre-pandemic period, as shown in Figure 12.

Afterwards, the content of the blogs was evaluated by uniformly trained experts and the posts were categorised into two themes: 1. the need for and feelings about public spaces for "companion animals" and 2. the experience of owning a companion animal. Finally, following semi-structured in-depth interviews, three central themes were identified: 1. changes in time and space for dog walking, 2. changes in motivations and attitudes towards dog walking, and 3. crisis management of pets during epidemics. Based on these three themes and the results of the analysis above, we continue in the next section of the discussion with a further analysis of the changing perceptions and use of public and pseudo-public spaces for "companion animals" in the city.

## 4. Discussion

The pandemic has profoundly changed public perceptions and usage of public and pseudo-public spaces for "companion animals". After reading all the comments, combined with the results of the big data analysis and the results of the in-depth interviews, referring to the COM-B behavioural model developed by Michie et al. [36], we conclude that the impact of the pandemic on the dynamics of direct interaction between humans and their pets with "companion animal" public and pseudo-public spaces can be summarised in three distinct but not mutually exclusive pathways: changes in opportunity, changes in capacity, and changes in motivation, as in Figure 14. Of course, the intensity and direction of these three pathways may vary considerably depending on demographic, regional, and national socioeconomic, political, cultural, and social factors and environmental factors, and potentially relevant changes in severity and response to the pandemic.

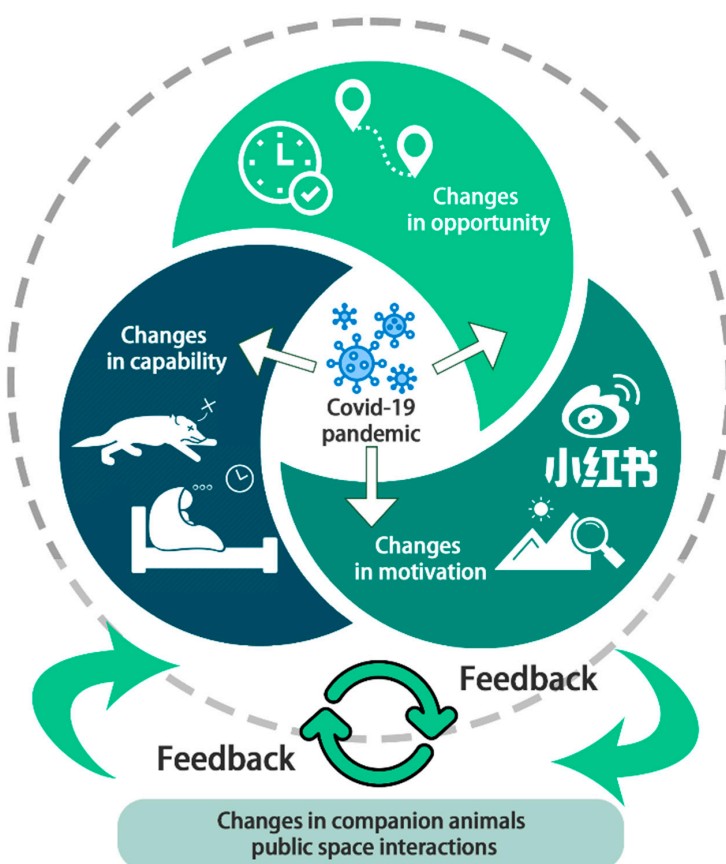

**Figure 14.** A conceptual framework of pandemic influences on the direct interaction of humans and their pets with "companion animals" public and pseudo-public spaces.

(1) Changes in opportunities related to the factors that facilitate or make possible spatial interactions with "companion animal" space, including the amount of interaction, the duration of contact, and how they interact. Firstly, regarding the number of interactions and the timing of contact, the pandemic had both positive and negative effects on humans and their pets with the "companion animal" space. On the positive side, the use of "companion animal" spaces increased for some residents during the pandemic. Survey data showed over 2000 blog posts expressing more time and similar sentiments during the pandemic, perhaps due to the use of teleworking during the pandemic increasing the time available for other activities for some residents [37]. As one user, "buzz Lightyear's wife", posted: "*I've spent a lot more time with my pals and had more opportunities to walk my dog this year because of the pandemic. It has probably been the most life-affirming year I have had.*" However, there is also a negative side to this, with less access to "companion animal" space for the wider public. During the pandemic, many places, such as shopping centres, community spaces, and major urban parks, were closed to reduce infection rates [38,39]. For low-risk and above-containment areas, working from home for long periods of time under the epidemic, people become distant from each other [40–42] and dogs' physical and mental health and well-being are seriously affected by reduced range of motion, less exercise, and less dog-to-dog communication [43]. As a result, the survey data show that, compared to the pre-pandemic period, the word frequency statistics show that words such as "Tickets", "Location", "Parking", and "Free", which express the strong demand of dog owners for "companion animal" public spaces, are still at the top of the list, while the frequency of activities such as "Camping", "BBQ", "Picnic", and "Blowing wind" and so on has also increased significantly, as shown in Figures 4 and 5. The reduced access to "companion animal" spaces may, therefore, be one of the main drivers of the increased demand for such green spaces during the pandemic. For example, one user named "Ollie with long eyelashes" excitedly posted about her desire: "*The closure was lifted at 12 pm, so I quickly asked my friends to take my pups out for a run, and I was very happy to have just a shallow 3-min run*".

During our in-depth interviews, interviewee 01 told us that, although some parks and green spaces were off-limits to dogs before the epidemic, in practice, the relevant no-dog rule existed in name only and, in some cases, it was possible to slip in and walk dogs when park guards were not looking. There was only a sign indicating the ban, and the associated penalties were almost non-existent. However, during the epidemic control period, almost all respondents mentioned that they found that even city parks near their residential areas were no longer allowed to walk their dogs, and that regulations were much stricter than before the epidemic. At the same time, the monitoring of dog walking in street green areas and green areas attached to buildings has also become very strict. Overall, respondents generally felt that, while, before the epidemic, it was mainly a subjective desire to take their pets to "companion animal" public spaces, after the epidemic, it was forced to become an immediate necessity, and the frequency and time spent walking dogs was significantly reduced. As a result of the epidemic, they rarely met other dog walkers, and thus many respondents reported that their and their dogs' social needs were not being met and that they were more depressed than before the epidemic. At the same time, respondent 04 mentioned that her dog walking behaviour and the way she used the public space for "companion animals" was very much dependent on the "dog walking culture" in the small area she visited, for example, whether dogs are walked on leashes, whether owners can follow their pets onto lawns, etc, which the big data cannot measure. After the epidemic, however, this dynamic use of "companion animal" public spaces has, to some extent, disappeared, as people are not allowed to gather in large numbers. In conclusion, the pandemic has increased the amount of time available to some people, but the amount of public space available for "companion animal" has decreased due to the closure and, with the normalisation of the pandemic, the construction and operation of urban public space will face new challenges. The next focus of research will be the balance between planning and design, management and operation, and policy implementation.

In addition, the pandemic may not only change the amount of "companion animal" space a person has access to, but also the way they interact with "companion animal" space. For urban green spaces, the closure of many major urban parks may have increased people's use of nearby natural environments and reduced their access to natural environments away from where they live. However, this is not absolute. For example, Figure 11 show that the general use of "companion animal" spaces by Beijing residents for several years has shifted from a concentration in built-up areas and city parks in the main urban areas before the pandemic to areas such as country parks outside of the city. We can see from this that, under the normal state of the pandemic, people are more inclined to go to suburban green spaces than before. For example, "Zhao Erdog loves chillies" posted: *"Under the cherished sun, I took my dog for a walk in this small wood. The city is becoming more and more green, but there are fewer and fewer places for the dogs to roam. Even without the towering walls, the walls built by the human heart are unbreakable. Do we have to take our dogs into the mountains to have a place in the valley? Alas! For now, let's just walk our dogs and cherish them."* For the built-up areas of the city, as parks and other green spaces are not the first places visited by pets and their owners, they are most often used in built-up areas [44]; usually, pet owners walk their dogs underneath their neighbourhoods or on the streets without a clear destination and, naturally, they do not need to make a point to record it. Thus, the main types of sites in our marker data are scenic and public green spaces, while built-up areas are mainly pseudo-public spaces, as shown in Figure 12. Pseudo-public spaces were more heavily regulated during the pandemic, so the number of geographic markers recommended was low.

In the in-depth interviews, the majority of respondents said that they went to pseudo-public spaces, such as shopping centres and plazas to walk their dogs prior to the epidemic in order to meet their own related needs and to take their dogs for a walk. However, there were some exceptions; for example, interviewee 12 mentioned going to private pet parks specifically for their dogs to give them an amusement-park-like experience, while some respondents said they went for grooming and maintenance and to show off their dogs and give their dogs a taste of "busy city life". However, during the epidemic, there were more risks to be avoided due to the ever-changing policies, and some respondents even said they did not walk their dogs in pseudo-public spaces at all after the outbreak. Unsurprisingly, the big data analysis points to a clear downward trend in the proportion of pseudo-public space. Although we cannot confirm whether this downward trend is due to the pandemic or privatisation, or a combination of both, it also reveals another problem: the use of "companion animal" public space in pseudo-public spaces decline in status, and those located within the open urban green space on the outskirts of the city stand out during the pandemic outbreak. How Beijing's economy and urban planning will respond to this change in spatial usage and its consequences will be an essential consideration for future urbanisation.

(2) Competence is the psychological and physical ability of pets and their owners to interact with "companion animal" public spaces. Firstly, on a physical level, it is clear that, if a person is infected with COVID-19, the pet will also be placed in temporary quarantine by the pet facility, which will stop their use of the "companion animal" space. However, for pets in pandemic areas, in addition to being taken away for isolation, there are also many one-size-fits-all policies, such as the one in Beijing's Daxing district, where a week-long dog-catching spree and forced "innocuity treatment" has caused discontent among dog owners. A netizen named "Christa-meng" commented on the news of the pet quarantine: *"Can we pay attention to the recent dog arrests in Daxing District, with police cars squatting in front of the district in the early hours of the morning? Why not go straight to the breeding source if they are not allowed to breed? It is not fair to euthanize a fur baby raised for seven or eight years outside the Fifth Ring Road because it was taken away with a single word!"* "A small milk dumpling said, *"This may cause some people who don't really love dogs to abandon them and turn them into stray dogs. It adds to the burden on society."* These comments quickly sparked outrage as the public began to expose various extreme COVID-19 prevention measures. Statistics show that negative sentiment increased by 10.24%. A careful reading of the blog post reveals three

main sources of negative comments; the first is dissatisfaction with the local community's one-size-fits-all policy of compulsory confiscation and "innocuity treatment" disposal of pets on health grounds, the second is dissatisfaction with the increasing difficulty in finding places to walk pets, and the third is dissatisfaction with the extreme negativity against pets in society, such as the extreme comments about directly linking pets to COVID-19.

In addition, in the in-depth interview, interviewee 06 told us that he had never travelled with his dog again afterwards because he had a need to transport his pets but, also, because the epidemic quarantine delayed a lot of time during the transport process and his own pets died of starvation and thirst as a result. At the same time, many interviewees also mentioned the complete lack of measures taken by the government during the epidemic, which was basically people-oriented, with dogs being treated as accessories to people. Some respondents said that they were forced to separate from their dogs because of the quarantine and could leave them with neighbours who had dogs or pet shops, while others said that there were no measures in place to deal with the situation and that their pets were forced to starve to death. It is also worth noting that the government has been lax in the management of dogs, as the epidemic is more loosely controlled in rural Beijing. As a result, survey respondents in rural areas reported that they were largely unaffected or minimally affected. A few respondents said that dogs were never taken out of their compounds, so, naturally, the impact was minimal. Overall, however, the cruel treatment of companion animals and the abandonment and even killing of animals to prevent humans from being infected in the above incidents reflect a strong anthropocentrism. Animals are reduced to resources for human growth when anthropocentrism is strong, and there is no ethical analysis of whether human demands and aspirations are appropriate. [45]. Emerging human-to-human infectious illnesses are seen as public health emergencies that solely endanger human health due to species barriers; hence, most matching emergency plans only offer treatment and refuge for people, while excluding the companion animals that share their homes. Although the central government and local authorities in China have begun to actively correct excessive pandemic prevention measures by issuing circulars to educate the public on animal protection, there are still no contingency plans proposed for the arrangement of relevant spaces.

On a psychological level, our sentiment analysis survey of the public opinion data sample showed a slight increase in overall negative sentiment, as the uncertainty and fear associated with the outbreak, as well as the massive blockade and economic recession, may have led to increased symptoms of anxiety, depression, post-traumatic stress disorder, and other forms of psychological disorders in the general population, even if people were not infected with COVID-19. At the same time, pets may experience behavioural changes due to changes in owners and the outside environment [46,47], such as frequent barking or fear of noise and the inability to be left alone in the home [48]. Such changes may facilitate the possibility of spatial interaction with "companion animals". As a result, the frequency of activity words has increased significantly in the survey data. For example, "Kafka Chou Chou" says: "*Every time the city was closed during the pandemic, I tried to take my dog Bao out to play, hiding her in the trunk every time, and then letting her sit in the car after the pandemic checkpoint, because she was not used to seeing people outside. Then depression usually stayed by itself, hiding under the table for half a day, and it was only afterwards that she was taken out for walks every day for about 3 months that she slowly got better*". Thus, we can also see from this that changes in ability sometimes lead to changes in motivation and, ultimately, to changes in behaviour and that there is a complex inter-relationship between the three drivers of the behavioural model.

(3) Motivation is a process in the human brain that motivates and guides behaviour, and the spread of the COVID-19 disease may have significantly altered people's motivation to interact with "companion animals" public space. In addition to the changes mentioned above in motivation due to changes in the available time, opportunities and accessibility, and physiological and psychological changes caused by the pandemic, during the pandemic, many users recommend "companion animal" spaces through geolocation marking and

the creation of various tags, such as "#Dangers of having a dog without a walk during a pandemic", created an information ecosystem defined by an unprecedented amount of data that profoundly influenced other users' motivations and behaviours [49,50]. For example, "small bok choy" said: "*I've always seen online that it's good to walk your dog here, so I stopped by today to take a look. I don't know, it's pretty big, love it, love it, and my boyfriend played in it all afternoon, tomorrow I'll bring Fu (dog's name) to experience it*". In our in-depth interviews, our respondents said that local residents in Beijing chose places to walk their dogs before the epidemic mainly out of local and lived experience, either by experiencing them first or by seeing people walking their dogs on the road and then recording them before taking them for a walk, or, in some cases, by recommendations from friends and family or by looking at tips on the internet. However, after the epidemic and, therefore, the ready change in policy, there was a lot of reliance on recommendations from the internet. However, this is not without its exceptions, as some respondents said that, in order to save time, they tended to go to familiar places rather than those recommended on the internet when there was a suitable space to walk their dog, and that, if they really wanted to walk their dog for a break, they basically chose to go to the more remote suburban green areas.

In addition, on the one hand, we can see from the pattern of changes in high-frequency words in Table 1, Figures 4 and 5, such as the significant increase in the frequency of activity words, that the positive attitudes of pet owners toward the public space of "companion animal" increased during the outbreak. On the other hand, however, the overall public's negative attitudes toward companion animals also increased significantly, even to pathological fear [7], which partly contributed to the statistical increase in the proportion of negative sentiments posted by pet owners during the pandemic. Although the two groups, pet owners and the wider public, partially overlap, the conflict between them exacerbates the social perception of multi-species conflict, leading to a variety of vicious conflicts [7] that are detrimental to the sustainability of the "companion animal" public space. We argue that social media plays a vital role during a pandemic, influencing the motivations and behaviours of users. Government interventions in the "companion animal" public space based on social media theories of critical knowledge, attitudes, intentions, and behaviour change could play a significant role. A shift from self-regulation of social media platforms to government intervention may be a positive step.

## 5. Conclusions

This paper examines the changing perceptions and use of "companion animals" in public and pseudo-public spaces in the city during the COVID-19 pandemic, using Beijing, China, as an example. The study was based on a Python web crawler that collected relevant text and comment data from Chinese social media Weibo and Xiaohongshu, followed by natural language (NLP) analysis, spatial analysis, and content analysis of the pre- and post-pandemic social media data, and, finally, semi-structured in-depth interviews were conducted. The big data revealed statistically significant differences in the perception and use of "companion animal" public spaces and pseudo-public spaces before and after the outbreak. Referring to the COM-B behavioural model developed by Michie et al., we conclude that the impact of the pandemic on the dynamics of direct interaction between humans and their pets with "companion animal" public and pseudo-public spaces can be attributed to three pathways: changes in opportunity, changes in ability, and changes in motivation, with complex inter-relationships between the three drivers. We found that the pandemic has acted as a mirror and catalyst to expose the multi-species coexistence of humans and animals in China's cities. The pandemic has increased the amount of time available to some people but reduced the amount of public space available for "companion animals" due to the pandemic's closure. As pandemics become normalised, the operation of public spaces related to "companion animals" in cities will face new challenges, and the balance between planning, design, management, operation, and policy implementation will be the next focus of research. In addition, due to species barriers, emerging human-to-human infectious diseases are perceived as public health crises that threaten only human

health. Therefore, most corresponding emergency plans only provide treatment and shelter for humans while neglecting the companion animals that live with them, especially in "companion animal" public spaces. There is still a lack of contingency planning by the Chinese central government and local authorities to arrange these spaces. In addition, the survey showed that the use of "companion animal" public spaces in pseudo-public spaces declined, while those in open urban green spaces on the outskirts of the city stood out after the outbreak, creating new requirements for the future development of Beijing's urban green space system and related policies. At the same time, pet owners' perceptions and behaviours regarding "companion animal" public spaces are heavily influenced by social media during pandemics, and the potential for abandonment and cross-species infections resulting from the acceptance of negative information about them can make companion animals a new public safety hazard. However, discussions about "companion animal" public spaces on social media are not actively regulated and appropriately intervened in, in some cases, leaving a very conflicted and confusing impression and stirring up group antagonism. We argue that there could be a shift from self-regulation on social media platforms to government intervention and joint efforts between government and social media to advocacy and support for sustainable animal ethics practices to better respond to the crisis.

Humans and companion animals merge, constitute, and permeate each other in everyday urban spaces. More research is needed to determine how to more thoroughly analyse the more complex relationships between "companion animal" public spaces and urban spaces and how these relationships are expressed in social media. The future of public space is uncertain due to the normalisation of pandemics, and more research is needed from different parts of the world.

**Author Contributions:** Conceptualisation, H.C. and W.D.; methodology, W.D.; software, H.C.; validation, W.D. and H.C.; formal analysis, H.C.; investigation, H.C.; resources, W.D.; data curation, H.C.; writing—original draft preparation, H.C.; writing—review and editing, W.D.; visualisation, H.C.; supervision, W.D.; project administration, W.D.; funding acquisition, W.D. All authors have read and agreed to the published version of the manuscript.

**Funding:** This research was funded by fundamental research funds for central universities (2018ZY10) and National Natural Science Foundation of China, project No. 51508024.

**Institutional Review Board Statement:** Ethical review and approval were waived for this study, due to involving no more than minimal risk.

**Informed Consent Statement:** Informed consent was obtained from all subjects involved in the study.

**Data Availability Statement:** Data supporting reported results can be found by contacting the authors.

**Acknowledgments:** I would like to thank Associate Professor Duan Wei for his strong support for this research work. Also, I would like to thank Haoning Liu and Shaojie Wang for their great contributions for this study.

**Conflicts of Interest:** The authors declare no conflict of interest.

# Appendix A

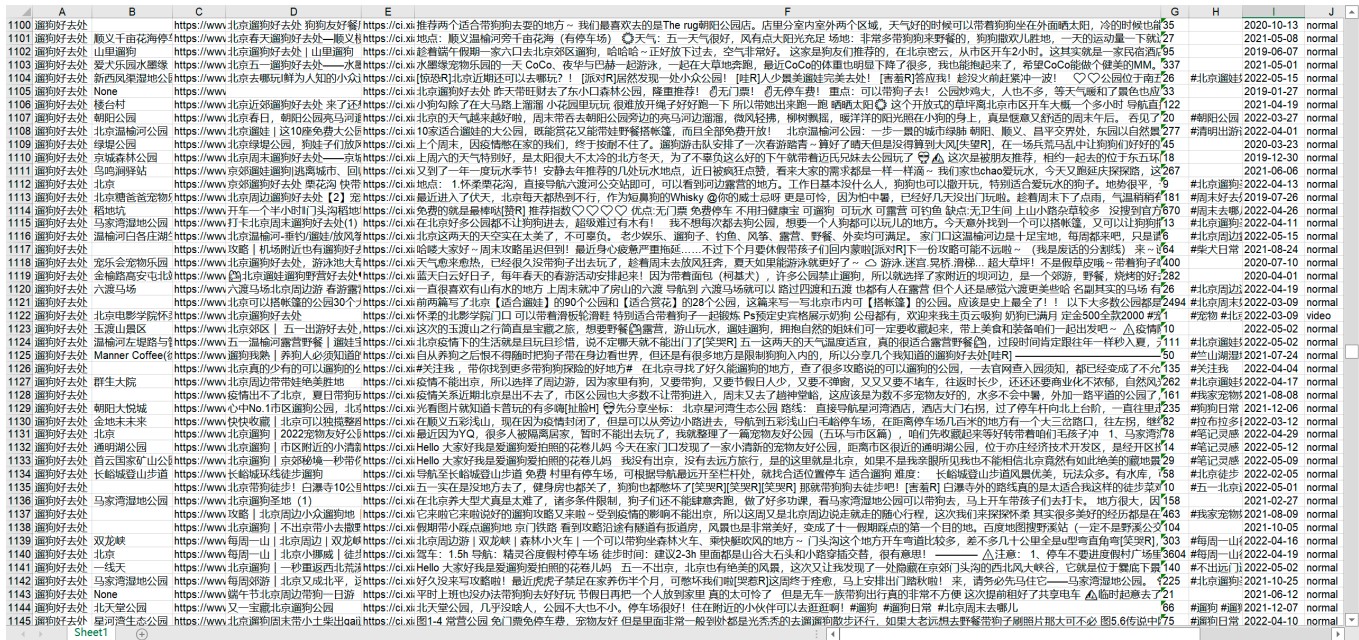

**Figure A1.** Some of the original text posted by Xiaohongshu users crawled using a web crawler.

**Figure A2.** Results of the sentiment analysis based on the *Chinese Sentiment Vocabulary Ontology Library*.

## Broad question set for the interview：

### Part 1 - Basic information

- What is your age? Gender? Level of education (tertiary or higher, below tertiary)? Employment status (employed, unemployed) and occupation? Household income (A. Below RMB 5,000 B. RMB 5,000-10,000 C. Above RMB 10,000)? Location of residence (urban, suburban, rural)? Self-rated physical condition (healthy, unhealthy)?

- How many years have you had the dog (how old is the dog)? Number? What breed? Size of dog?

### Part2—General and material impact

- What were the main places where dogs were usually walked before the outbreak? What were the main types? What was the frequency and duration? What would you say is particularly convenient or inconvenient about the place where you walk your dog?

- What are the main places where dogs are usually walked after the epidemic? What were the main types? What was the frequency and duration? What would you say is particularly convenient or inconvenient about the place where you walk your dog?

- Do you think time spent with dogs and walking them has increased or decreased since the outbreak and why?

- How were your dogs handled during the closure? Did the government give any support or issue any policy on this?

- How do you usually find a suitable place to walk your dog? By what means? Do you have any online recommendations for places to walk your dog?

- Have you tried private pet parks and taking your dog for grooming and maintenance? Has the frequency of visits to relevant places changed since the outbreak? Why?

- Have you tried walking your dog in shopping centres and plazas, such as Sanlitun and Outlets, and has the frequency of visits to these places changed since the outbreak? Why?

### Part3—Psychological and spiritual influences

- Has there been any significant change in your dog's mood or behaviour since the outbreak? Has there been any change in the number of interactions with you?

- Has the mindset and reasons for walking dogs changed or are there any concerns about walking dogs after the outbreak compared to before the outbreak? Specifically?

- Do you think the quality of life for the dogs went up or down during the epidemic especially during the closure? What were the overall benefits and drawbacks?

**Figure A3.** Broad question set for the semi-structured in-depth interviews.

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
