# Peer review of "Changing Perceptions and Uses of “Companion Animal” Public and Pseudo-Public Spaces in Cities during COVID-19 Pandemic: The Case of Beijing"

_land, doi:10.3390/land11091475_

Round 1

Reviewer 1 Report

Manuscript: "Changing Perceptions and Uses of 'Companion Animal' Public and Pseudo-Public Spaces in Cities during COVID-19 Pandemic: The Case of Beijing" contains interdisciplinary research aimed at revealing changes in human-companion animal relations under COVID-19 forced isolation, in the context of analyzing the use of public spaces.

It should be noted the originality of the research methods used in the work, which are described in sufficient detail in the manuscript. They are based on Big Data methodology, and the analysis of publications on the Internet to determine the most frequent words to build a map of clouds of perceptual words, as sensors of processes occurring during the COVID-19 outbreak and before it. Such method allows to cover a big number of respondents for analysis, but it differs essentially from traditional sociological researches (questionnaires, polls, etc.). Statistical processing of results shows high accuracy and reliability.

Fig. 1 and Fig. 2 are not understandable for most readers of the journal, they should be translated, but in this case they will lose their meaning. It would be better to remove them, especially since a detailed description is given in the text, and Figures 4 and 5 present word clouds in the language established for manuscripts.

The results of the study are extensive, addressing social issues, particularly the effects of isolation measures on both humans and pets (dogs). Issues of placement of companion animal facilities during and after an epidemiological outbreak. 

An analysis of changes in the perception and use of public and pseudo-public spaces for "companion animals" in the city is detailed and well illustrated. The results presented here are useful for the development of the city's master plan, showing the differences in the use of natural areas and pseudo-public areas during the outbreak and normal times.

The conclusions are sound and follow from the text.

The bibliography contains a sufficient number of articles on the topic under consideration.

The manuscript corresponds to the topics of the journal, contains new approaches to solving urgent problems of sustainable urban development, is recommended for publication in Special Issue: Effects of the COVID-19 Pandemic on the Use and Perception of Urban Green Space.

Author Response

Dear reviewer,

Thanks very much for taking the time to review this manuscript. We really appreciate all your valuable comments and suggestions! Please find our carefully provided responses below and our corrections in the re-submitted files.

Point 1: Fig. 1 and Fig. 2 are not understandable for most readers of the journal, they should be translated, but in this case, they will lose their meaning. It would be better to remove them, especially since a detailed description is given in the text, and Figures 4 and 5 present word clouds in the language established for manuscripts.

Response 1: After careful consideration, we feel that it is more appropriate to include Figures 1 and 2 as raw data in the appendix, which will also provide the reader with additional perspectives on our study. At the same time, we have added semi-structured in-depth interviews to acquire a more rigorous dimension. Due to the length of the revision,

(1) For the description of the in-depth interview methodology, please see lines 270 to 310 on page 6 of the revised version of the manuscript.

(2) For the conclusions drawn from the in-depth interviews, please see lines 405 to 410 on page 13, lines 472 to 492 and 520 to 528 on page 19, lines 559 to 584 on page 20, and lines 620 to 631 on page 21.

(3) For the basic problem setting, please see Figure A3 in Appendix A.

(4) The remaining changes are necessary to refine the logic of the thesis and are detailed in lines 96 to 98 on page 2, lines 125 to 131 and 140 to 201 on page 3, line 209 on page 5, and lines 308 to 310 on page 7, lines 396 to 398 on page 14, line 433 on page 17 and lines 653 to 654 on page 22.

Thank you again for all your suggestions and thank you so much for your approval of our work.

Reviewer 2 Report

Abstract

            The scope of this research paper has been pointed out well enough, in the abstract of the piece, which is, to engage in an analysis of companion animals and changing perceptions towards the same, in the public spaces as well as pseudo public spaces in the country of China. The location of the study, that is, the fact that it is concentrated on the Chinese city of Beijing, and that the nature of the study is a case analysis is mentioned in the abstract as well. The rationale for the study and key aims and objectives of the paper, are mentioned in the research abstract.

Introduction                         

            The introductory section of this research paper has been written in quite an elaborate manner. Here too, there is an extensive background which has been provided by the authors of the paper about why it is that companion animals in public spaces and pseudo public spaces is being studied and what is it that can be gained from undertake such a project in the first place. The possible conclusions of the paper are also hinted at, in the introductory section itself.

Literature Review

            The literature review which has been conducted by the authors of this piece is also quite extensive in nature. There are a number, of papers which have been reviewed, that have companion animal in urban spaces, as the main theme of the study, and after the review of all, of the papers was undertaken, the gaps in the review that could be detected have been mentioned as well. In doing so, the rationale for conducting the present study is something that has been reiterated further. There is plenty of care that has been taken by the authors of the piece in selecting the articles for review and analysis, ensuring that the review of literature has been conducted as appropriately as possible.

Methodology

            It is important to make note of the fact that this is a study which has engaged in the qualitative and quantitative data on the presence of companion animals in urban spaces of Beijing, with such data having been retrieved from two popular social media platforms in China. Attempts have been made through the textual and numerical analysis of this data to see how common it is to see companion animals in urban spaces of China, especially in the public spaces and what it is that their significance is.

Results

            The results of the study have been depicted quite graphically. There are maps, charts, diagrams, and visual images which have been used on the part of the authors, to indicate what the social media data analysis conveys.

Conclusion

            The paper concludes that there is indeed a transformation that can be detected in pseudo public and public spaces in China, when it comes to companion animals but that this change or transformation is still quite nascent.

Comments

- The researchers of this study have done a thorough job of evaluating qualitative and quantitative data, to arrive at the necessary conclusions. 

- However, more in-depth primary research in the form of surveys, interviews and focus discussions need to be conducted, for such a study to acquire a more rigorous dimension.

- Also for a large subject as that need to enrich the references.

- Some of the figures are not clear enough (visualy)

Author Response

Dear reviewer,

Thanks very much for taking your time to review this manuscript. We really appreciate all your valuable comments and suggestions which have significantly raised the quality of the manuscript and have enabled us to improve the manuscript! Please find our carefully provided responses below and our corrections in the re-submitted files.

Point 1: However, more in-depth primary research in the form of surveys, interviews and focus discussions need to be conducted, for such a study to acquire a more rigorous dimension.

Response 1: Thank you very much for your comments. After careful consideration, we have decided to use semi-structured in-depth interviews for a more in-depth preliminary study. It can also compensate to some extent for the shortcomings of big data surveys and can be cross-checked with the findings of big data. Due to the length of the revision,

(1) For the description of the in-depth interview methodology, please see lines 270 to 310 on page 6 of the revised version of the manuscript.

(2) For the conclusions drawn from the in-depth interviews, please see lines 405 to 410 on page 13, lines 472 to 492 and 520 to 528 on page 19, lines 559 to 584 on page 20, and lines 620 to 631 on page 21.

(3) For the basic problem setting, please see Figure A3 in Appendix A.

(4) The remaining changes are necessary to refine the logic of the thesis and are detailed in lines 96 to 98 on page 2, lines 125 to 131 and 140 to 201 on page 3, line 209 on page 5, and lines 308 to 310 on page 7, lines 396 to 398 on page 14, line 433 on page 17 and lines 653 to 654 on page 22.

Point 2: Some of the figures are not clear enough (visualy)

Response 2: (1) We have revised the Figure 1 Framework diagram of research ideas on page 7 of the manuscript, lines 308-310, by adding an in-depth interview process and adding main outcomes at the bottom of the diagram which gives the reader a better understanding of our research thinking and methodology. (2) We have also re-uploaded Figure 13. (a) Coupling of user geographic markers and planning maps (before the outbreak); (b) Coupling of user geographic markers and planning maps (during the outbreak) located on pages 16 and 17 to ensure resolution and clarity of viewing.

Point 3: The cited references relevant to the research must be improved. Also for a large subject as that need to enrich the references.

Response 3: We have carefully rechecked the references one by one, finding a small number of errors in citation numbers due to typographical problems, as well as replacing what we consider to be inappropriate examples of citation presentation and inadequate citations, all of which have been corrected. The modifications are as follows. Page 1 lines 34-37, page 2 lines 44-49, 54-56, 58-62, 80, page 5, line219.

Thank you again for all your suggestions and thank you so much for your approval of our work.
